# *Transform Once*
# Efficient Operator Learning in Frequency Domain

**Michael Poli**\*
Stanford University
DiffeqML

**Stefano Massaroli**\*
Mila
DiffeqML

**Federico Berto**\*
KAIST
DiffeqML

**Jinykoo Park**
KAIST

**Tri Dao**
Stanford University

**Christopher Ré**
Stanford University

**Stefano Ermon**
Stanford University
CZ Biohub

## Abstract

Spectral analysis provides one of the most effective paradigms for information-preserving dimensionality reduction, as simple descriptions of naturally occurring *signals* are often obtained via few terms of periodic basis functions. In this work, we study deep neural networks designed to harness the structure in frequency domain for efficient learning of long-range correlations in space or time: frequency-domain models (FDMs). Existing FDMs are based on complex-valued transforms i.e. *Fourier Transforms* (FT), and layers that perform computation on the spectrum and input data separately. This design introduces considerable computational overhead: for each layer, a forward and inverse FT. Instead, this work introduces a blueprint for frequency domain learning through a single transform: *transform once* (T1). To enable efficient, direct learning in the frequency domain we derive a variance preserving weight initialization scheme and investigate methods for frequency selection in reduced-order FDMs. Our results noticeably streamline the design process of FDMs, pruning redundant transforms, and leading to speedups of 3 x to 10 x that increase with data resolution and model size. We perform extensive experiments on learning the solution operator of spatio-temporal dynamics, including incompressible Navier-Stokes, turbulent flows around airfoils and high-resolution video of smoke. T1 models improve on the test performance of FDMs while requiring significantly less computation (5 hours instead of 32 for our large-scale experiment), with over $20\%$ reduction in predictive error across tasks.

## 1 Introduction

*Nature uses only the longest threads to weave her patterns, so that each small piece of her fabric reveals the organization of the entire tapestry. (Feynman, 1965)*

Naturally occurring *signals* are often sparse when projected on periodic basis functions (Strang, 1999). Central to recently-introduced instances of frequency-domain neural operators (Li et al., 2020; Tran et al., 2021), which we refer to as *frequency-domain models* (FDMs), is the idea of learning to modify specific frequency components of inputs to obtain a desired output in data space. With a hierarchical structure that blends learned transformations on frequency domain coefficients with regular convolutions, FDMs are able to effectively approximate global, long-range dependencies in higher resolution signals without requiring prohibitively deep architectures.

---

\*Equal contribution authors. Contact email: `poli@stanford.edu`.

36th Conference on Neural Information Processing Systems (NeurIPS 2022).

Yet, existing FDMs suffer from several drawbacks:

1. **Slow inference:** every layer of an FDM performs a forward and inverse frequency domain transform, introducing a considerable computational overhead.

2. **Expensive parameter scaling:** each layer of typical FDMs (Li et al., 2020) performs a long convolution over the inputs by parametrizing it in frequency domain, which scales poorly in the signal resolution.

3. **Incompatibility:** parameter initialization schemes and layers devised to learn directly in data space can be highly suboptimal when introduced without modifications to FDMs.

Despite attempts to improve performance (Gupta et al., 2021; Tran et al., 2021), scaling FDMs to larger data resolutions and model sizes remains fundamentally challenging[2].

In this work, we start by posing the question:

*To reap the benefits of learning on frequency domain representations, is it necessary to construct hierarchical deep models that perform forward and inverse frequency transforms at each layer?*

We provide the answer in *Transform Once* (T1), a model that builds representations directly in frequency domain, after a *single* forward transform. Each aspect of T1 addresses a specific limitation of existing FDMs:

1. **Fast:** by performing a single forward transform and optimizing directly on frequency domain coefficients of target data, T1 iterations are *at least* 3x to 10x faster. When scaling to larger models and higher resolutions, the relative speedups increase as the overhead of each transform grows.

2. **Favourable scaling:** T1 employs a single real-valued transform, which we observe to stabilize training and finetuning of deep networks in frequency domain.

3. **Enhanced compatibility:** removing redundant transforms streamlines the design space for T1 architectures compared to existing FDMs, allowing direct introduction of optimized layers developed for other applications e.g. UNets (Ronneberger et al., 2015).

In § 2.1 we provide a (short) history on frequency domain approaches in deep learning, followed by background on FDMs § 2.3. In § 3, we describe how to train T1 directly in frequency domain and motivate the choice of DCT, in § 3.1 we discuss how to choose modes of reduced-order FDMs and in § 3.2 we introduce a simple variance-preserving weight initialization scheme for all FDMs. Finally, in § 4 we evaluate T1 on a suite of benchmarks related to learning solution operators for a variety of dynamics: incompressible Navier-Stokes, flow around different airfoil geometries, and high-resolution videos of turbulent smoke (Eckert et al., 2019).

Across tasks, T1 is $3\times$ to $10\times$ faster and reduces predictive errors by $20\%$ on average. Training T1 models on high resolution videos (600 x 1062) of turbulent dynamics is significantly faster, requiring 5 hours instead of 32 hours (FNOs) for the same number of iterations.

## 2 Related Work and Background

### 2.1 Learning and Frequency Domain: A Short History

Links between frequency-domain signal processing and neural network architectures have been explored for decades, starting with the original CNN designs (Fukushima and Miyake, 1982). Mathieu et al. (2013); Rippel et al. (2015) proposed replacing convolutions in pixel space with element-wise multiplications in Fourier domain. In the context of learning to solve *partial differential equations* (PDEs), *Fourier Neural Operators* (FNOs) (Li et al., 2020) popularized the state-of-the-art FDM layer structure: forward transform → learned layer → inverse transform. Similar architectures had been previously proposed for generic image classification tasks in (Pratt et al., 2017; Chi et al., 2020). Modifications to the basic FNO recipe are provided in (Tran et al., 2021; Guibas et al., 2021; Wen et al., 2022). A frequency domain representation of convolutional weights has also been used

---

[2]Existing methods to overcome this limitation avoid the frequency domain of inputs, instead introducing an intermediate patch embedding step (Guibas et al., 2021; Pathak et al., 2022).

for model compression (Chen et al., 2016). Fourier features of input *domains* and periodic activation functions play important roles in deep implicit representations (Sitzmann et al., 2020; Dupont et al., 2021; Poli et al., 2022) and general-purpose models (Jaegle et al., 2021).

## 2.2  Learning to Solve Differential Equations

A variety of deep learning approaches have been developed to solve differential equations: neural operators and physics-informed networks (Long et al., 2018; Raissi et al., 2019; Lu et al., 2019; Karniadakis et al., 2021), specialized architectures (Wang et al., 2020; Lienen and Günnemann, 2022), hybrid neural-numerical methods (Poli et al., 2020; Kochkov et al., 2021; Mathiesen et al., 2022; Berto et al., 2022), and FDMs (Li et al., 2020; Tran et al., 2021), the focus of this work.

## 2.3  Frequency-Domain Models

Let $\mathcal{D}_n$ ($n$-space) to be the set of real-valued discrete signals[3] of resolution $N$. Our objective is to develop efficient neural networks to process discrete signals $x \in \mathcal{D}_n$,
$$x_0, x_1, \ldots, x_{N-1}, \quad x_n \in \mathbb{R}.$$
We define a layer of FDMs mapping $x$ to an output signal $\hat{y} \in \mathcal{D}_n$ as the structured operator:

$$
\begin{aligned}
X &= \mathcal{T}(x) && \text{Forward Transform} \\
\hat{X} &= f_\theta(X) && \text{Learned Map} \\
\hat{x} &= \mathcal{T}^{-1}(\hat{X}) && \text{Inverse Transform} \\
\hat{y} &= \hat{x} + g(x) && \text{Residual}
\end{aligned}
\tag{1}
$$

where $\mathcal{T}$ is an orthogonal (possibly *complex*) linear operator. We denote the $\mathcal{T}$-transformed $n$-space with $\mathcal{D}_k$ ($k$-space) so that $\mathcal{T} : \mathcal{D}_n \to \mathcal{D}_k$. Typically, we assume $\mathcal{T}$ to be a *Fourier-type* transform[4] (Oppenheim, 1999, Chapter 8) so that the $k$-space corresponds to the *frequency domain* and its elements form the *spectrum* of the input signal $x$.

The learned parametric map $f_\theta : \mathcal{D}_k \to \mathcal{D}_k$ is the stem of a FDM layer: it maps the $k$-space into itself and is typically chosen to be rank-deficient in the linear case, e.g. $f_\theta(X) = S_m^\top A(\theta) S_m X$, $A(\theta) \in \mathbb{C}^{m \times m}$ ($m \leq N$). The matrix $S_m \in \mathbb{R}^{n \times m}$ selects $m$ desired elements of $X$, setting the rest to zero. In the case of frequency domain transforms, this allows (1) to preserve or modify only specific frequencies of the input signal $x$.

Residual connections or residual convolutions $g$ (Li et al., 2020; Wen et al., 2022) are optionally added to reintroduce frequency components filtered by $S_m$. A FDM mixes global transformations applied to coefficients of the chosen transform to local transformations $g$ i.e. convolutions with finite kernel sizes. To ensure that such models can approximate generic nonlinear functions, nonlinear activations are introduced after each inverse transform.

**Fourier Neural Operators**  Layers of the form (1) appear in recent FDMs such as *Fourier Neural Operators* (FNOs) (Li et al., 2020) and variants (Tran et al., 2021; Guibas et al., 2021; Wen et al., 2022).

In example, an FNO is recovered from (1) by letting $\mathcal{T}$ be a *Discrete Fourier Transform* (DFT)
$$\hat{x} = \mathcal{T}^{-1} \circ f_\theta \circ \mathcal{T}(x) = W^* S_m^\top A(\theta) S_m W x$$
where $W \in \mathbb{C}^{N \times N}$ is the standard $N$-dimensional DFT matrix and $W^*$ its conjugate transpose. The *Discrete Fourier Transforms* (DFTs) is a natural choice of $\mathcal{T}$ as it can be computed in $O(N \log N)$ via *Fast Fourier Transform* (FFT) algorithms (Oppenheim, 1999, Chapter 9.2).

We identify two major limitations of FDMs in the form (1); each layer performs $\mathcal{T}$ and $\mathcal{T}^{-1}$ and DFTs are complex-valued, resulting in overheads and a restriction of the design space for $f_\theta(X)$.

With T1, we aim to develop an FDM that does not require more than a single $\mathcal{T}$, while preserving or improving on predictive accuracy. Ideally, the transform in T1 should be (1) real-valued, to avoid restrictions in the design space of the architecture and thus retain compatibility with existing

---

[3]For clarity of exposition, models and algorithms proposed in the paper are introduced without loss of generality for one-dimensional scalar signals (i.e. $\mathcal{D}_n \equiv \mathbb{R}^n$).

[4]e.g. *discrete Fourier transform* (DFT), *discrete cosine transform* (DCT), etc.

pretrained models, (2) universal, to allow the representation of target signals, and (3) approximately sparse or structured, to allow dimensionality reduction.

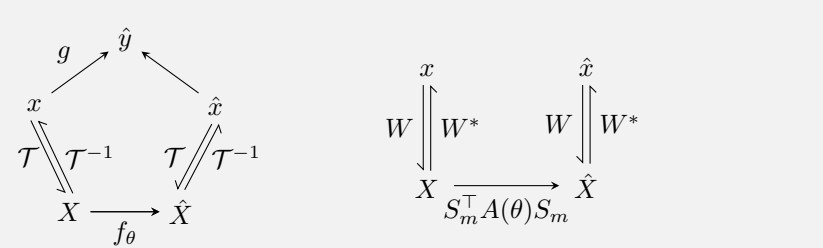

Commutative diagrams for FDM layers (1) and linear FNOs (frequency domain part).

## 3 Transform Once: The `T1` Recipe

With `T1`, we introduce major modifications to the way FDMs are designed and optimized. In particular, `T1` is defined, inferred and trained directly in the frequency domain with only a **single** direct transform required to process data. Hence follows the name: *transform once* (`T1`).

**Direct learning in the frequency domain**  Consider two signals $x \in \mathcal{D}_n$, $y \in \mathcal{D}_n$ and suppose there exists a function $\varphi : \mathcal{D}_n \to \mathcal{D}_n$ mapping $x$ to $y$, i.e.

$$y = \varphi(x).$$

Then, there must also exist another function $\psi : \mathcal{D}_k \to \mathcal{D}_k$ that relates the spectra of the two signals, i.e. $Y = \psi(X)$ being $X = \mathcal{T}(x)$ and $Y = \mathcal{T}(y)$. In particular,

$$\varphi(x) = \mathcal{T}^{-1} \circ \psi \circ \mathcal{T}(x) \iff \mathcal{T} \circ \varphi(x) = \psi \circ \mathcal{T}(x)$$

It follows that, from a learning perspective, we can aim to approximate $\psi$ directly in the $k$-space rather than $\varphi$ in the $n$-space. To do so, we define a learnable parametric function $f_\theta : \mathcal{D}_k \to \mathcal{D}_k$ and train it to minimize the approximation error $J_\theta$ of the output signal spectrum $Y$ in the $k$-space. Given a distribution $p(x)$ of input signals, `T1` is characterized by the following nonlinear program

$$
\begin{aligned}
\min_\theta \quad & \mathbb{E}_{x,y}\left[\|\mathcal{T}(y) - \hat{Y}\|\right] \\
\text{subject to} \quad & \hat{Y} = f_\theta \circ \mathcal{T}(x) \\
& x \sim p(x) \\
& y = \varphi(x)
\end{aligned}
\tag{2}
$$

If $\mathcal{T}$ is a DFT, the above turns out to be a close approximation (or equivalent, depending on the function class of $f_\theta$) to the minimization of $\|y - \hat{y}\|$ in $n$-space by the *Parseval-Plancherel identity*.

**Theorem 3.1** (Parseval-Plancherel Identity (Stein and Shakarchi, 2011, pp. 223) )**.** *Let $\mathcal{T}$ be the normalized DFT. Given a signal $v \in \mathcal{D}_n$ and its transform $V = \mathcal{T}(v)$, it holds $\|v\| = \|V\|$.*

This result also applies to any other norm-preserving transform $\mathcal{T}$, e.g. a normalized type-II DCT (Oppenheim, 1999, pp. 679). For the linear transforms considered in this work, $\mathcal{T}(x) = Wx$, $W \in \mathbb{C}^{N \times N}$, condition for Th. 3.1 to hold is $W$ to be orthonormal, i.e. $W^*W = \mathbb{I}$.

Note that `T1` retains, in principle, the same universal approximation properties of FNOs (Kovachki et al., 2021) as $f_\theta$ is allowed to operate on the entirety of the input spectrum. Given enough capacity, $f_\theta$ can arbitrarily approximate $\psi$, implicitly reconstructing $\varphi$ via $\mathcal{T}^{-1} \circ f_\theta \circ \mathcal{T}$.

**Speedup measurements**  We provide a concrete example of the effect of pruning redundant transforms on computational costs. We measure wall-clock inference time speedups of depth $d$ `T1`

$$\text{T1}(x) := f_d \circ \cdots \circ f_2 \circ f_1 \circ \mathcal{T}(x)$$

over an equivalent depth $d$ FNO with layers (1). The only difference concerns the application of transforms between layers.

Fig. 3.1 provides the speedups on two-dimensional signals: on the left, we fix model depth $d = 6$ and investigate the scaling in signal width (i.e. number of channels) and signal resolution. On the

right, we fix signal width to be 32 and visualize the interaction of model depth and signal resolution. For common experimental settings e.g. resolutions of 64 or 128, 6 layers and width 32, T1 is at least 10 x faster than other FDMs. It will later be shown (§ 4) that T1 also preserves or improves on predictive accuracy of other FDMs across tasks.

When T1 is not preceded by online preprocessing steps for inputs $x$, such as other neural networks or randomized data augmentations, the transform on $\mathcal{T}(x)$ can be done once on the dataset, amortizing the cost over training epochs, and increasing the speed of T1 further.

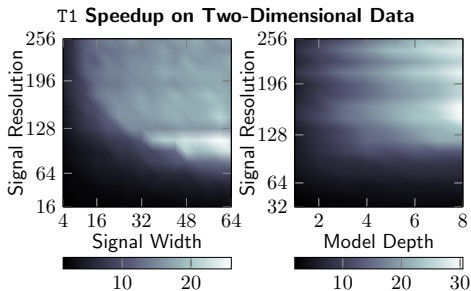

**Choosing the right transform**   The transform $\mathcal{T}$ in T1 is chosen to be in the class of *Discrete Cosine Transforms* (DCTs) (Ahmed et al., 1974; Strang, 1999), in particular the normalized DCT-II, which can also be computed $\mathcal{O}(N \log N)$ via FFTs (Makhoul, 1980). DCTs provide effective representations of smooth continuous signals (Trefethen, 2019) and are the backbone of modern lossy compression methods for digital images.

Figure 3.1: Speedup in a forward pass of T1 over FNOs sharing the same transform $\mathcal{T}$ (DFT) on two-dimensional signals of increasing resolution. The speedup for a given configuration (point on the plane) is shown as background color gradient. The improvement grows with signal width, resolution and model depth.

Although other transforms are available, we empirically observe DCT-based T1 to perform best in our experiments. This phenomenon can be explained by the sparsity and energy distribution properties of the transformed spaces, an intrinsic property of the specific dataset and chosen transform. This is in line with results of classic signal processing and compression literature. Particularly, DCT features are known to have a higher energy compaction than their DFT counterparts in a variety of domains, from natural images (Yaroslavsky, 2014) to audio signals (Soon et al., 1998). Energy compaction is often the decisive factor in choosing a transform for downstream tasks.

Letting $\mathcal{T}$ be a real-valued transform in T1 architectures preserves compatibility between $f_\theta$ and existing architectures e.g., models pre-trained on natural image datasets.

### 3.1   Reduced-Order T1 Model and Irreducible Loss Bound

We seek to leverage structure induced in $\mathcal{D}_k$ by $\mathcal{T}$. To this end we allow T1, similarly to (1), to modify specific elements of $X$ and consequently trasform only certain frequency components of $x$ (and $y$).

The reduced-order T1 model is designed to operate only on $m < N$ elements (selected by $S_m \in \mathbb{R}^{N \times m}$) of the input $k$-space, i.e. on a *reduced $k$-space* $\mathcal{D}_m \equiv \mathbb{R}^m$ of lower dimension. Thus, we can employ a smaller neural network $\gamma_\theta : \mathcal{D}_m \to \mathcal{D}_m$ for mapping $S_m X$ to the corresponding $m$ elements $S_m Y$ of the output $k$-space. Thus, training involves a truncated objective that compares predictions with elements in the output signal spectrum also selected by $S_m$:

$$
\begin{aligned}
\min_\theta \quad & \mathbb{E}_{x,y}\left[\|S_m \circ \mathcal{T}(y) - \hat{Y}\|\right] \\
\text{subject to} \quad & \hat{Y} = \gamma_\theta \circ S_m \circ \mathcal{T}(x) \\
& x \sim p(x) \\
& y = \varphi(x)
\end{aligned}
\tag{3}
$$

**How to choose modes in reduced-order FDMs** We now detail some intuitions and heuristic to choose which modes $k_0, \ldots, k_{m-1}$ should be kept to maximize the information content in the truncated spectrum. For this reason, we evaluate the *irreducible* loss arising from discarding some $N - m$ modes. We recall that the (reduced) $k$-space training objective $J_\theta(X, Y)$ reads as

$$J_\theta(X, Y) = \|S_m Y - \hat{Y}\| = \sum_{l=1}^{m} |Y_{k_l} - \gamma_{\theta, k_l} \circ S_m(X)|,$$

since only the first $m$ predicted output modes $\hat{Y}_{k_1}, \ldots, \hat{Y}_{k_m}$ can be compared to $Y_k$. We then consider the total loss $L_\theta$ of the approximation task, including the $N - m$ elements of the output $k$-space discarded by our model, i.e.

$$L_\theta(X, Y) = \|Y - S_m^\top \hat{Y}\| = \underbrace{\sum_{l=0}^{m-1} |Y_{k_l} - \gamma_{\theta, k_l} \circ S_m(X)|}_{J_\theta(X,Y)} + \underbrace{\sum_{k=m}^{N-1} |Y_k - 0|}_{R_o(Y)}.$$

It follows that the overall loss $L_\theta$ is higher than T1's training objective $J_\theta$, i.e. $L_\theta = J_\theta + R_o > J_\theta$, whilst $R_o$ represents the *irreducible* residual loss due to truncation of the predictions $\hat{Y}_k$.

**Optimal mode selection in auto-encoding T1** In case $Y = X$, i.e. the reduced-order T1 is tasked with reconstructing the input spectrum, the optimal modes minimizing the irreducible loss are the ones with highest magnitude. This can be formalized as follows.

**Proposition 3.1** (Top-$m$ modes minimize the irreducible loss). *Let $Y = X$ (reconstruction task). Then the choice $k_0, \ldots, k_{m-1} = \mathrm{top}_k(m) \ |X_k|$ minimizes the irreducible loss term $R_o$.*

This means that if the spectrum of $X$ is monotonically decreasing in magnitude, then low-pass filtering is the optimal mode selection.

**Corollary 3.1** (Low pass filtering is optimal for monotonic spectrum). *If $|X_k|$ is monotonically decreasing in $k$, then the choice $k_0, \ldots, k_{m-1} = 0, \ldots, m - 1$ minimizes the residual $R_o$.*

However, spectra in practical datasets are commonly non-monotonic e.g., the spectrum of solutions of chaotic or turbulent systems (Dumont and Brumer, 1988). We show an example in Fig. 3.2.

**Mode selection criteria in general tasks** When $Y \neq X$ and the task is a general prediction task, the simple $\mathrm{top}\, m$ analysis is not optimal. Nonetheless, given a dataset of input-output signals it is still possible to perform an *a priori* analysis on $R_o$ to inform the choice of the best modes to keep.

Often, we empirically observe the irreducible error $R_o$ for reduced-order T1 to be smaller than for non-reduced-order FDMs i.e $R_o < \sum_{k=m}^{K-1} \|Y_k - \mathcal{T}_k(\hat{y})\|$ with layers of type (1)[5].

We also note that the reachable component $J_\theta$ of the objective cannot always be minimized to zero regardless of the approximation power of $\gamma_\theta$. For each $k < m$, $S_m$ discards $N - m$ frequency

---

[5]See Fig. 4.1 and Appendix B for experimental evidence in support of this phenomenon.

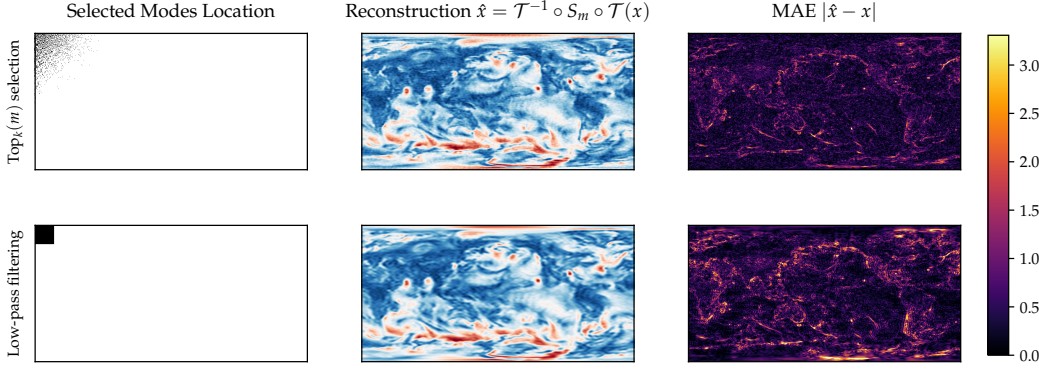

Figure 3.2: Reconstructions after low-pass filtering (first $m$ modes) **[Bottom]** or top-$m$ selection **[Top]** of ERA5 (Hersbach et al., 2020) climate data. The non-monotonic structure of the spectrum implies more accurate reconstructions can be obtained with top-$m$ selection.

components of the input signal which, if different than zero, likely contain the necessary information to approximate $\psi_k(X)$ exactly. Specifically, the irreducible lower bound on $J_\theta$ should depend on "how much" the output's $m$ frequency components depend on the discarded $N-m$ input's elements.

A rough quantification of such bound can be obtained by inspecting the mismatch between the gradients of $\psi_k - \gamma_{\theta,k} \circ S_m$ with respect to $X$. In particular, it holds

$$\sum_{j=0}^{N-1} \left| \frac{\partial \psi_k(X)}{\partial X_j} - \frac{\partial \gamma_{\theta,k}(S_m X)}{\partial X_j} \right| = \sum_{j=0}^{m-1} \left| \frac{\partial \psi_k(X)}{\partial X_j} - \frac{\partial \gamma_{\theta,k}(S_m X)}{\partial X_j} \right| + \sum_{j=m}^{N-1} \left| \frac{\partial \psi_k(X)}{\partial X_j} \right|,$$

Unless $\partial_{X_j}\psi_k(X) = 0$ holds for all $j = m, \ldots, N-1$ and $k = 0, 1, \ldots, N-1$ i.e. no dependency of the ground truth map in $k$-space on the truncated elements, there will be an irreducible overall gradient mismatch and thus a nonzero $J_\theta$.

## 3.2 Weight Initialization for Reduced-Order FDMs

FDMs (Li et al., 2020; Tran et al., 2021; Wen et al., 2022) opt for a standard Xavier-like (Glorot and Bengio, 2010) initialization distribution that takes into account the input channels $c$ to a layer i.e. $\mathcal{N}(0, \frac{1}{c})$. However, well-known variance preserving properties of Xavier schemes do not hold for FDM layers truncating $N - m$ elements of the $k$-space. Notably, Xavier schemes do not scale the variance of the weight initialization distribution based on the number of elements $m$ kept after truncation of the spectrum performed by $f_\theta$, leading to the *collapse* of the outputs to zero.

To avoid this issue in T1 and other FDMs, we develop a simple *variance-preserving* (vp) that introduces a variance scaling factor based on $m$ and the class of transform.

**Theorem 3.2** (Variance Preserving (vp) Initialization). *Let $\hat{x} = W^* S_m^\top A S_m W x$ be a $k$-space reduced-order layer and $W$ is a normalized DCT-II transform. If $x \in \mathbb{R}^N$ is a random vector with*
$$\mathbb{E}[x] = \mathbb{0}, \quad \mathbb{V}[x] = \sigma^2 \mathbb{I}.$$
*Then,*
$$A_{ij} \sim \mathcal{N}\left(0, \frac{N}{m^2}\right) \Rightarrow \mathbb{V}[\hat{x}] = \mathbb{V}[x].$$

We report the proof in Appendix A, including some considerations for specific forms of $f_\theta$.

**Corollary 3.2** (vp initialization for DFTs). *Under the assumptions of Theorem 3.2, if $W$ is a normalized DFT matrix we have* $\mathrm{Re}(A_{ij}), \mathrm{Im}(A_{ij}) \sim \mathcal{N}(0, \frac{N}{2m^2}) \Rightarrow \mathbb{V}[\hat{x}] = \mathbb{V}[x]$.

The collapse phenomenon is empirically shown in Fig. 3.3 for $m = 24$, comparing a single layer of FNO and FFNO (with Xavier initialization) with FNO equipped with the proposed vp scheme. Under the assumptions of Corollary 3.2, we sample $A$ and compute empirical variances of $\hat{x} = W^* S_m^\top A(\theta) S_m W x$ for several finite batches of input signals $x$. We repeat the experiment for signals of different lengths $N$. The vp scheme preserves unitary variances whereas the other layers concentrate output variances towards zero at a rate that grows with $N - m$.

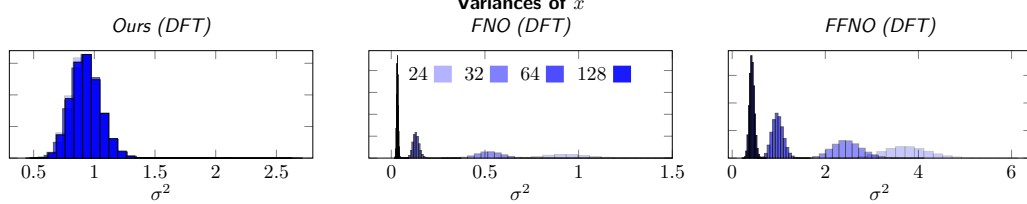

Figure 3.3: Output variance histogram in layer outputs $\hat{x} = W_m^* S_m^\top A(\theta) S_m W_N$, for a finite sample of inputs $x$ and a single sample of $\theta$. Color indicates signal resolution.

When the learned frequency-domain transformation $f_\theta$ is obtained, instead of the single low-rank linear layer $f_\theta = A(\theta) S_m X$, as the composition of several layers, preserving variances can be achieved by applying the vp scheme only to the first layer. For some variants of FDMs e.g. FNO that truncate the spectrum at each layer, vp initialization should instead be applied to all.

# 4  Experiments

We validate `T1` on learning to approximate solution operators of dynamical systems from images.

- In § 4.1, we apply `T1` on the standard task of learning solution operators for incompressible Navier-Stokes, comparing against other FDMs. In § 4.1.1 we perform a series of ablation experiments on each ingredient of the `T1` recipe, including weight initialization and architecture. In § 4.1.2 we provide scaling laws.

- In § 4.2 we deal with fluid-solid interaction dynamics in the form of higher resolution images (128). We consider turbulent flows around varying airfoil geometries, benchmarking against current SOTA (Thuerey et al., 2020).

- In § 4.3 we show how the computational efficiency of `T1` allows learning on unwieldy data without downsampling or building low-resolution meshes. We consider learning on high-resolution video ($600 \times 1062$) capturing the turbulent dynamics of smoke (Eckert et al., 2019).

Configuration and model details are reported in the supplementary material. The code is available at `https://github.com/DiffEqML/kairos`. *Weights & Biases* (`wandb`) (Biewald, 2020) logs of results are provided.

## 4.1  Incompressible Navier-Stokes

We show that `T1` matches or outperforms SOTA FDMs with less computation on the standard incompressible Navier-Stokes benchmark. Losses are reported in $n$-space (signal space) for comparison.

**Setup**  We consider two-dimensional Navier-Stokes equations for incompressible fluid in vorticity form as described in (Li et al., 2020). Given a dataset of initial conditions, we train all models to approximate the solution operator at time $50$ seconds for high viscosity ($\nu = 1e^{-3}$) and at time $15$ for lower viscosity ($\nu = 1e^{-4}$). As a metric, we report *normalized mean squared error* (N-MSE). Both initial condition as well as solution are provided as images of resolution $64$.

We include as baseline established FDMs, such as Fourier Neural Operators (FNOs) (Li et al., 2020) and *Factorized Fourier Neural Operators* (FFNOs) (Tran et al., 2021). We indicate with the suffix *vp* models that employ the proposed variance preserving initialization scheme. All models truncate to $m = 24$, except FFNOs to $m = 32$.

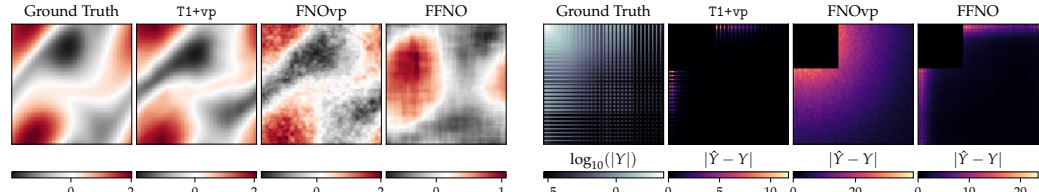

Figure 4.1: **[Left]** Direct predictions at $T = 50s$ on high viscosity Navier-Stokes. **[Right]** Ground-truth spectrum and absolute errors in $k$-space (DCT-II). Despite predicting only the first $m = 24$ elements, reduced-order `T1` models produce smaller errors even in other regions of the $k$-space.

**Results**  We perform 20 training runs for each model and report mean and standard deviation in Table 4.1. `T1` reduces solution error w.r.t FNOs by over $20\%$ and FFNOs by over $40\%$. A single forward pass of `T1` models is on average 2x faster than FNO and 10x than FFNOs. We note that FFNOs

| Method | Param. (M) | Size (MB) | Step (ms) | high $\nu$ | low $\nu$ |
|---|---|---|---|---|---|
| FFNO (Tran et al., 2021) | 8.9 | 35 | 294 | 0.997±0.003 | 1.016±0.010 |
| FNO (Li et al., 2020) | 14.2 | 56 | 31 | 0.379±0.006 | 0.328±0.004 |
| FNOvp | 14.2 | 56 | 32 | 0.351±0.003 | 0.315±0.006 |
| T1+vp | 10.2 | 40 | 19 | **0.257**±0.007 | **0.240**±0.004 |

Table 4.1: Benchmarks on incompressible Navier-Stokes. Direct long-range prediction errors (N-MSE) in $n$-space (signal space) of different models.

are designed to share parameters between layers and thus require deeper architectures – and slower, due to more transforms. In particular, training time (500 epochs) for `T1` is cut to 20 minutes down from 40 of FNOs, matching the model speedup. Finally, we report an improvement in performance for FNOs with parameters initialized following our proposed scheme (FNOvp). Fig. 4.1 provides sample predictions in $n$-space (left) to contextualize the task, in addition to prediction errors in frequency domain (right). Despite being a reduced order model with $m = 24$, `T1+vp` produces smaller errors on truncated $k$-space elements ($k > m$) compared to FNOvp and FFNO.

### 4.1.1 Ablations on weight scheme and architecture

We repeat the previous experiment and report prediction errors for four variants of `T1`: same architecture and weight initialization scheme as FNOs (`T1`), `T1` with our proposed vp scheme (`T1vp`), a reduced-order variant with $k$-space model $f_\theta$ defined as a UNet architecture (`T1+`), and `T1+` with variance preserving scheme (`T1+vp`). The results in Table 4.2 provide empirical evidence in support of the vp scheme and its synergistic effect with the proposed architecture. In particular, combining vp scheme and UNet structure in frequency domain reduces error by half compared to the naive `T1` approach.

| Method | high $\nu$ | low $\nu$ |
|:---:|:---:|:---:|
| T1 | 0.491 | 0.449 |
| T1vp | 0.304 | 0.280 |
| T1+ | 0.295 | 0.260 |
| T1+vp | **0.257** | **0.240** |

Table 4.2: Ablation on the effect of the proposed weight initialization scheme and T1 architecture.

### 4.1.2 Scaling laws

We verify whether the reduction in predictive error of `T1` over neural operator baselines is preserved as the size of training dataset grows. We perform 10 training runs on the Navier-Stokes $\nu = 1e^{-4}$ experiment, each time with a larger dataset size, and report the scaling laws in Fig. 4.2. With additional data, the gaps in test errors narrow slightly, with noticeable improvements obtained by applying the vp scheme to both FNO and `T1+`.

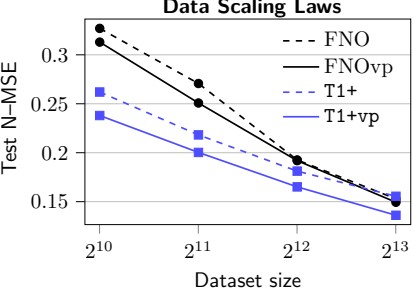

Figure 4.2: Scaling laws for N-MSE.

### 4.2 Flow Around Airfoils

We investigate the performance of `T1` in predicting steady-state solutions of flow around airfoils.

**Setup** We use data introduced in (Thuerey et al., 2020) in the form of 10000 training pairs of initial conditions, specifying freestream velocities and the airfoil mask, with the target steady-state velocity and pressure fields. This task introduces additional complexity in the form of higher resolution input images (128) and a full $k$-space due to the discontinuity in the field produced by the mask.

We compare a SOTA UNet architecture (DFPNet) introduced by (Thuerey et al., 2020) to FNOs and `T1` with vp initialization schemes. We perform a search on the most representative hyperparameters (detailed in the Appendix). Averages for 5 runs are reported in Table 4.3.

| Method | N-MSE | Time (hrs) |
|:---:|:---:|:---:|
| DFPNet | 0.023 | **1.3** |
| FNO | **0.020** | 6.0 |
| T1+vp | 0.024 | **1.3** |

Table 4.3: Test N-MSE and total training time on the flow around airfoil task.

**Results** All models are able to accurately predict steady-state solutions for different airfoils with small normalized errors. Test N-MSE is comparable as all models are within a single standard deviation. Training of `T1` is as fast as DFPNets (Thuerey et al., 2020) and as accurate as FNOs, as evidence of the applicability of `T1` to tasks with signals that are not band-limited (in this case due to the airfoil mask).

### 4.3 Turbulent Smoke

We investigate the performance of `T1` in predicting iterative rollouts from high-resolution video of real rising smoke plumes.

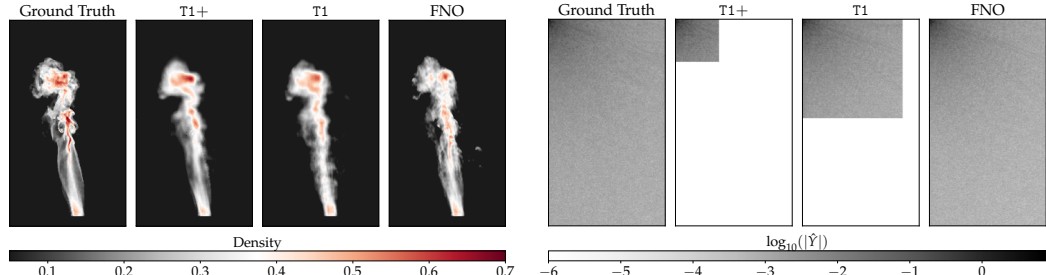

Figure 4.3: **[Left]** 10-step rollout predictions on ScalarFlow. FNOs produce high-frequency, non-physical artifacts and accumulate error more rapidly in time compared to `T1` models **[Right]** Log-absolute values of predictions in $k$-space (DCT-II). Although `T1` is limited to $m = 512$ and `T1+` to $m = 224$ $k$-space elements, the predictions are overall more physically accurate in $n$-space.

**Setup**   We use the ScalarFlow dataset introduced in (Eckert et al., 2019) consisting of 104 sequences of 150 frames each collected from video recordings of rising hot smoke plumes. The dataset consists of raw video data at high-resolution (600 × 1062) collected at 60 fps. This task scales up complexity by involving real-world high-definition data, capturing highly-turbulent dynamics. We perform rollouts iteratively based on previous predictions: all models are trained on 3-step rollouts and evaluated over 10-steps extrapolation to test their generalization in time. We compare FNOs against `T1`, `T1+` and `T1+vp` of similar model sizes after performing a search on most representative hyperparameters (Appendix B).

**Results**   Fig. 4.3 provides a sample rollout of different model predictions in $k$-space (DCT-II). `T1+vp` accumulates smaller errors over the rollout and is less prone to generating non-physical artifacts by performing prediction only on a subset of the $k$-space (Table 4.4). Notably, `T1` and `T1+` are $4\times$ to $7\times$ faster, providing a reduction in training time from $32.4$ hours to $4.7$. Appendix B includes additional visualizations, including averaged prediction errors on $k$-space.

| Method | N-MSE | Time (hrs) |
|--------|-------|------------|
| FNO    | 0.232 | 32.4 |
| T1     | 0.239 | 8.1 |
| T1+    | 0.256 | 4.7 |
| T1+vp  | **0.228** | **4.7** |

Table 4.4: Test 10-steps rollout $n$-space prediction errors (N-MSE) and total training time on the ScalarFlow dataset.

## 5   Conclusion

We present a streamlined class of *frequency domain models* (FDM): *Transform Once* (`T1`). `T1` models are optimized directly in frequency domain, after a single transform, and achieve similar or improved predictive performance at a fraction of the computational cost (3x to 10x speedups across tasks). Further, a simple truncation-aware weight initialization scheme is introduced and shown to improve the performance of `T1` and existing FDMs.

## Acknowledgments

This work is supported by NSF (1651565), AFOSR (FA95501910024), ARO (W911NF-21-1-0125), ONR, DOE, CZ Biohub, Sloan Fellowship and JSPS Kakenhi (21J14546).

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
