# OpenReview forum: "Transform Once: Efficient Operator Learning in Frequency Domain"
_NeurIPS.cc/2022/Conference — NeurIPS 2022 Accept_

### Official Review · Reviewer_Q4Qr · 2022-07-04

**Rating:** 8
**Confidence:** 4
**Soundness:** 4 excellent
**Presentation:** 4 excellent
**Contribution:** 3 good

**Summary:**

This paper proposes a frequency domain approach for deep learning that is very intuitively summarized in the title: the goal is to have a single transformation into the frequency domain, perform all necessary operations there, and then transform back. While not an overly expensive operation, it is an interesting goal to reduce the number of frequency domain operations. The paper arrives at this goal via truncation and a DCT.

The paper is very nicely written, and motivates the approach well. Frequency domain models (FDMs) are an active topic of research, and the paper thoroughly compares to popular variants, the "fourier neural operators" and a factorized version. It evaluates the proposed approach with a set of flow-centric set examples. This limits the scope somewhat, but I personally found the examples very interesting, and I believe they cover a nice variety of interesting structures for learning problems.

The paper also comes with some theoretical results. They look convincing, but I have to admit I didnt check the proofs in detail.


**Questions:**

One open point I noticed was the introduction and later on omission of the T1+vp scheme. Section 4.1.1 first discussed its clear advantages, but the more complex examples in 4.2 and 4.3 seem to only use T1+. Can the authors explain this?


**Limitations:**

As mentioned above, I think the low-pass filtering due to the frequency truncation should be discussed more explicitly.


**Strengths And Weaknesses:**

In terms of main strengths, I think the paper proposes a very attractive idea. The presentation, derivation and discussion are great, I believe. The paper is clear, and very nice to read. The results and presented in a compelling way, and provide a detailed picture of the proposed approach. I think this is a clear strength of the paper.

The advantages of the resulting algorithm are probably a bit less than what one would hope for given the initial discussion of positive properties of the single transformation, but I think they're nonetheless substantial. I definitely found the results in terms of MSE errors and resource usage to be attractive steps forward.

On the side of weaknesses, the results can nonetheless be seen as somewhat underwhelming at times. I think especially the strong low-pass filtering of the frequency truncation is a central limitation of the approach. This is, e.g., visible in the turbulent smoke outputs (fig. 4.3). For this setup, more examples should be provided in the appendix, and I think the implications of inherent low-pass filtering (and the corresponding smoothing effects) should be discussed in the main paper. I hope the authors can also comment on this in the rebuttal, but for now I'm sure the authors could address this in a finalized version of their submission.

Overall, I would argue for accepting this paper to NeurIPS. It proposes an interesting variant of FDMs, thoroughly evaluates and discusses it. Even if some of the results "only" result in smoother outputs with similar MSE and faster runtime, I think the approach is suitable for presentation at NeurIPS as a variant that is highly attractive for some learning scenarios.

---

> ### Author Response · Authors · 2022-08-02
> **Discussion with Reviewer Q4Qr**
>
> Thank you for the thoughtful review and your detailed comments. We are excited to receive such positive feedback on soundness, presentation and contribution of T1! Please see some of our other responses for some additional results on generalization of T1 models to different resolutions, as well as additional details to motivate our claims of parameter-inefficiency of FNOs ([precise parameter counts](https://gist.github.com/anonymous-conf-sub/9faa4c8e24a13b0e8f78be91b6eef0e7)).
>
> > On the side of weaknesses, the results can nonetheless be seen as somewhat underwhelming at times. I think especially the strong low-pass filtering of the frequency truncation is a central limitation of the approach. This is, e.g., visible in the turbulent smoke outputs (fig. 4.3). For this setup, more examples should be provided in the appendix, and I think the implications of inherent low-pass filtering (and the corresponding smoothing effects) should be discussed in the main paper. I hope the authors can also comment on this in the rebuttal, but for now I'm sure the authors could address this in a finalized version of their submission.
>
> While the experiments in the first version on the manuscript followed the same approach *low-pass filtering* of FNOs (e.g. chosing the top-left $m$-by-$m$ block of $X$ in the 2D case) other approaches can certainly be developed to select the spectral components of the reduced-order model. In our formulation, this boils down to a particular choice of the operator $S_m$ which can be designed *a priori* or computed adaptively given $X$, i.e. $S_m = S_m(X)$. While a thorough investigation of the topic lies beyond the scope of this paper, we are happy to share a preliminary exploration of alternative mode selection methods. See `lowpass_vs_topk_pass.pdf` in our [anonymous gist](https://gist.github.com/anonymous-conf-sub/9faa4c8e24a13b0e8f78be91b6eef0e7), where we show ScalarFlow reconstruction quality metrics with basic low-pass selection for DCT modes compared to an alternative approach in which we select the top-k $m$ modes with highest *energy compaction* i.e. the modes that maximize $\sum_{k\in\text{selected modes}}X_k^2$.
>
> We are planning to include additional visualizations and comments on the importance of mode selection and its band-pass effect on predictions.
>
> > One open point I noticed was the introduction and later on omission of the T1+vp scheme. Can the authors explain this?
>
> Thank you for pointing this out! The first submission did not include metrics for T1s with different weight initialization schemes on DFP and ScalarFlow due to computation constraints. Showing convincing ablations for weight initialization requires a large number of training runs, which are expensive for large-scale ScalarFlow prediction.
>
> We have since been able to perform these additional experiments. We are excited to share a full ablation between T1, T1+, T1+vp (similar to Section 4.1.1) for the large-scale ScalarFlow task! The result confirms the observations made in 4.1.1. We include in the updated table for the ScalarFlow experiment the _PSNR_ (Peak Signal to Noise Ratio): we note that the reconstruction quality of the rollout for $\tt T1$ models also matches with the trend in N-MSE:
>
> | Method | N-MSE | PSNR | Time (hrs) |
> | -------- | -------- | -------- | -------- |
> | FNO     | 0.220 $\pm$ 0.002   | 27.93 $\pm$  0.08   | 32.4 |
> | $\tt T1$     | 0.214  $\pm$  0.002   | 28.04  $\pm$  0.11   | 8.1 |
> | $\tt T1+$     | 0.203 $\pm$  0.004    | 28.50  $\pm$  0.13   | 4.7 |
> | $\tt T1+vp$     | 0.199  $\pm$  0.005   | 28.54 $\pm$  0.22     | 4.7 |
>
> Mean and standard deviations are computed with $5$ runs. We are performing additional runs to finalize our metrics.

---

> > ### Comment · Reviewer_Q4Qr · 2022-08-07
> > **Rebuttal**
> >
> > I'd like to thank the authors for their comments and clarifications.
> >
> > I've also looked at the other reviews, and I don't share the viewpoint that the contributions aren't sufficient. I think even if the step forward is not huge, this work potentially could inspire other follow up work, and I'd still support accepting it.

---

> > > ### Author Response · Authors · 2022-08-09
> > > **Thank you for the feedback**
> > >
> > > Thank you for the response and valuable feedback. Please let us know if there are further questions to address.

---

### Official Review · Reviewer_42JU · 2022-07-05

**Rating:** 7
**Confidence:** 5
**Soundness:** 3 good
**Presentation:** 4 excellent
**Contribution:** 3 good

**Summary:**

The submission develops a new model architecture under the umbrella of _frequency-domain models_. These models are used to learn reduced order models of dynamical systems. The main development is to apply an integral transform (such as an FFT or DCT) once to the signal and then apply a chain of non-linear operations solely in the spectral domain before transforming back. This saves a lot of computation moving back and forth to the frequency domain at every layer.

The authors test their model on a number of dynamical systems modelling problems, showing superior performance in terms of normalized mean squared error and wall clock time. The paper is well written and I support acceptance.

**Questions:**

20-21: The original quote is from Feynman (1965)

39: What is meant be a "non-trivial number of channels"? Also, please provide an actual example of how you arrive at hundreds of millions of parameters? If you bandlimit the signals in the frequency domain, as performed in the original Fourier Neural Operator paper, I think the number of learnable parameters is significantly lower than this.

Sec 2.1: Regarding learning in the frequency domain, consider also the line of works in the equivariance literature, where the irreducible representations are used for filters.

89: I would be careful not to use the notation $\mathcal{D}_n$ and $\mathcal{D}_k$ for signal space and frequency space. It is very easy to assume that these are the same kind of space, just with a different dimensionality, when, for example, one may be a real space and the other a complex one.

100: For the Fourier neural operator, is $g$ not purely local; that is, it is a 1x1 filter?

110: How is the complex-valuedness of the layers related to parameter--inefficiency? Since signals are real, there is conjugate symmetry in the frequency domain, so while needing two real parameters to represent each complex parameter, we only need half the filter taps anyway.

- Please use equation numbers for all display-mode equations

186: What is the necessity for measuring the high frequency loss if your task is not reconstruction? $R_o$ will be non-zero, but I do not see why reconstruction error is a good measure of performance unless a downstream task is specified.

- One of the considerations for using FNO is that convolution with large kernels in the spatial domain becomes more efficient in the spectral domain, since the FFT diagonalizes convolution. With bandlimiting of the spectral representation of the kernel, this becomes very efficient. In this work, $f_\theta$ is left free-form and thus while adding capacity, does not lead to a computational complexity reduction necessarily. I'm guessing the main complexity reduction is in the fact that you do not need to transform back and forth from the spectral domain with every layer?

- Is the use of N-MSE the best performance metric for rollout quality? In Fig 4.3 it appears the T1+ loses a lot of fidelity compared to the FNO model baseline. This would not be captured by N-MSE, but by a spectral metric.

- Do this model exhibit generalization over resolution, like the original FNO models? It does not appear to me that this would happen.

**Ethics Review Area:**

["I don’t know"]

**Limitations:**

The authors do not address limitations of their work. This should be fixed if the draft is to published. Conversations about resolution independence, choice of integral transform, and restrictions to regular grids would be welcomed.

**Strengths And Weaknesses:**

+ 153: I like the insight that $\mathcal{T}$ can be amortized across training epochs.

+ Section 3.3: The weight initialization scheme for the reduce-order FDMs looks sound. It is nice that this is shown in the main body of the submission.

+ The experimental results look good. The experiments themselves appear to have been conducted properly (+1 for using multiple training runs).

+ I like the idea of transforming once to the spectral domain and then performing all computations there before transforming back to the spatial domain. It saves computation and thus allows one to deploy much more expressive models in the spectral domain, which is exactly what this submission does (they use a U-Net).

+ The submission is well written and nicely typeset.

- The authors have not included limitations of their work.

- Some of the claims about baselines (see questions) are a bit exaggerated---for instance, FNO does not use hundreds of millions of parameters (see Table 4.1) and these shown be toned down.

---

> ### Author Response · Authors · 2022-07-31
> **Discussion with Reviewer 42JU [1/3]**
>
> Thank you for the review and the detailed, thoughtful questions: we are thrilled to see such a positive reception!
>
> We include below some exciting new experiments on generalization of T1 models to different resolutions, as well as miscellaneous results to contextualize our claims of parameter-inefficiency of FNOs.
>
> > Some of the claims about baselines (see questions) are a bit exaggerated---for instance, FNO does not use hundreds of millions of parameters (see Table 4.1) and these should be toned down. [...] please provide an actual example of how you arrive at hundreds of millions of parameters? If you bandlimit the signals in the frequency domain, as performed in the original Fourier Neural Operator paper, I think the number of learnable parameters is significantly lower
>
> FNOs used for Navier-Stokes (resolution: $64$) are smaller, as verified in Table 4.1. The parameter-inefficiency of spectral convolution layers used by FNOs is more apparent when scaling to signals of higher resolution, and choosing to threshold at a larger number of modes $m$.
>
> Please see `parameters_fno_vs_t1.pdf` in this [anonymous gist](https://gist.github.com/anonymous-conf-sub/9faa4c8e24a13b0e8f78be91b6eef0e7) where we provide a full table of parameter counts at different number of modes $m$ and channel widths $c$. We note that we took band-limiting into account. Hopefully, these concrete examples will provide much-needed context to our claims! For a fair comparison, one should compare parameter counts at $m$ for T1+ and $\approx m / 2$ for FNOs due to conjugacy arguments of Fourier transforms on real signals (see for example our approximation error decay Figure  B.5: DCT-II and DFT naturally perform similarly, provided one accounts for conjugacy).
>
> We will tone down the claims in the introduction, clarifying that FNOs with "hundreds of millions of parameters" only occurs when attempting to scale up FDMs to longer signals. As you point out, not all FNOs are as large (for example our Navier Stokes FNOs).
>
> > Q: For the Fourier neural operator, is not purely local; that is, it is a 1x1 filter?
>
> In Fourier Neural Operators, $g$ is most often a $1$x$1$ convolution ([code example of $g$ in the original FNO](https://github.com/zongyi-li/fourier_neural_operator/blob/74b1572d4e02f215728b4aa5bf46374ed7daba06/fourier_2d.py#L95)). Other architectural blocks have been considered in the literature; for example, few blocks of a UNet operating in signal domain [1].
>
> [1] "U-FNO - an enhanced Fourier neural operator-based deep-learning model for multiphase flow"
>
> > Q: How is the complex-valuedness of the layers related to parameter--inefficiency? Since signals are real, there is conjugate symmetry in the frequency domain, so while needing two real parameters to represent each complex parameter, we only need half the filter taps anyway.
>
> This is an important point. In our revision, we plan to provide further context to our claim, as it seems to have been the cause of a (small) misunderstanding.  We report here a summary of our response to Reviewer kKUQ addressing a similar question:
>
> It is true that for a real signal $x$, conjugate symmetry can be exploited to reduce the number of operations required for learned transformation in frequency domain $\hat{X} = f_\theta(X)$. The empirical observations of DCT outperforming DFT in the tasks we considered could be explained by the sparsity and energy distribution properties of the transformed spaces, i.e. an intrinsic property of the specific dataset and a chosen transform. This is in line with results of classic signal processing and compression literature. Particularly, DCT features are known to have a better *energy compaction* (which roughly translates to *sparsity* in ML terms) than their DFT counterparts in a variety of domains, from natural images to audio signals. Energy compaction is often the decisive factor in choosing a transform for downstream tasks e.g., compression (JPEG relies on DCT).
>
> We provide additional experiments where we directly train deep FDMs in complex k-spaces (T1-DFT). T1-DFT is less stable during training and diverges, with gradient norms exploding after few training iterations. This leaves complex k-space FDMs to rely on single (or few) linear layers which are parameter-inefficient when compared against optimized, free-form architectures. Please see `parameters_fno_vs_t1.pdf` in this [anonymous gist](https://gist.github.com/anonymous-conf-sub/9faa4c8e24a13b0e8f78be91b6eef0e7) where we provide a full table of parameter counts at different number of modes $m$ and channel width $c$. For a fair comparison, one should evaluate parameter counts at $m$ for T1+ and $\approx m / 2$ for FNOs due to conjugacy arguments of DFTs on real signals.
>
> [2] "Fourier Neural Operator for Parametric Partial Differential Equations"

---

> > ### Author Response · Authors · 2022-07-31
> > **Discussion with Reviewer 42JU [2/3]**
> >
> > > Q: What is the necessity for measuring the high-frequency loss if your task is not reconstruction?  will be non-zero, but I do not see why reconstruction error is a good measure of performance unless a downstream task is specified
> >
> > Measuring "importance" of a spectral domain mode for different is, in our opinion, one of the most stimulating areas of further investigation for reduced-order T1 models. In this work, we focus on a reconstruction loss on the (target) signal, which is natural for the prediction tasks considered. Note how we do not specify the nature of our target $Y$ in spectral domain! In particular, we propose to choose the $m$ modes such that the irreducible loss $R_o$ is minimal ($R_o$ represents a lower bound on the error achievable by the model, see e.g Figure B.5, where we show how close trained T1 models get to this theoretical lower bound).
> >
> > > One of the considerations for using FNO is that convolution with large kernels in the spatial domain becomes more efficient in the spectral domain, since the FFT diagonalizes convolution. With bandlimiting of the spectral representation of the kernel, this becomes very efficient.
> >
> > T1 models retain all computational advantages of performing computation in the spectral domain. T1s, as other FDMs, can (if needed) truncate or filter certain spectral domain coefficients, reducing the effective dimensionality of the data. Learned spectral layers for T1s are similarly advantageous, in that they can mimic large kernel, signal domain convolutions by local computation in spectral domain.
> >
> > > In this work, $f$ is left free-form and thus while adding capacity, does not lead to a computational complexity reduction necessarily. I'm guessing the main complexity reduction is in the fact that you do not need to transform back and forth from the spectral domain with every layer?
> >
> > That's spot on! Other than retaining all the advantages mentioned above, T1s are computationally more lightweight as they do not rely on redundant transforms. Free-form layers in spectral domain can also be much more parameter efficient than spectral convolutions of FNOs, as discussed above (see T1+ vs FNO parameter counts discussed previously). All of these factors play an important role in the speedups observed between T1s and other FDMs.
> >
> > > Is the use of N-MSE the best performance metric for rollout quality? In Fig 4.3 it appears the T1+ loses a lot of fidelity compared to the FNO model baseline. This would not be captured by N-MSE, but by a spectral metric.
> >
> > To further contextualize our results, we include in the updated table for the ScalarFlow experiment the _PSNR_ (Peak Signal to Noise Ratio): we note that reconstruction quality of the rollout for $\tt T1$ models matches the trend in N-MSE. We plan to include additional rollout visualizations in our revision, pointing out of the high-frequency predictions of FNOs are misleading in terms of fidelity. We have also ablated the proposed $vp$ weight initialization on the ScalarFlow task, showing a consistent trend with the results of Section 4.1.1:
> >
> > | Method | N-MSE | PSNR | Time (hrs) |
> > | -------- | -------- | -------- | -------- |
> > | FNO     | 0.220  $\pm$ 0.002   | 27.93  $\pm$ 0.08   | 32.4 |
> > | $\tt T1$     | 0.214  $\pm$ 0.002   | 28.04  $\pm$ 0.11   | 8.1 |
> > | $\tt T1+$     | 0.203 $\pm$ 0.004    | 28.50  $\pm$ 0.13   | 4.7 |
> > | $\tt T1+vp$     | 0.199  $\pm$ 0.005   | 28.54 $\pm$ 0.22     | 4.7 |
> >
> > We welcome further suggestions on metrics of interest that might give us further insights on model performance!

---

> > > ### Author Response · Authors · 2022-07-31
> > > **Discussion with Reviewer 42JU [3/3]**
> > >
> > > > Q: Do this model exhibit generalization over resolution, like the original FNO models? It does not appear to me that this would happen.
> > >
> > > Great question! We are happy to share additional results showing that T1 models can indeed generalize over resolution. We train FNO and T1+ models on Navier-Stokes data of resolution $64$ x $64$, and test them without finetuning on simulation of different (higher) resolutions:
> > >
> > > N-MSE across different resolutions. Navier-Stokes, viscosity $1e^{-4}$, $T=15$:
> > >
> > > | Method | 64 | 128 | 256 | 512 | 2056
> > > | -------- | -------- | -------- | -------- | ------ |  ------
> > > | FNO     | 0.308     | 0.301     | 0.302 | OOM  | OOM
> > > | T1+vp     | 0.285  | 0.286  | 0.287  | 0.286  | 0.285  |
> > >
> > > Time (s) to process a single test batch of data at different resolutions:
> > >
> > > | Method | 64 | 128 | 256 | 512 | 2056
> > > | -------- | -------- | -------- | -------- | ------ |  ------
> > > | FNO     | 0.049     | 0.049     | 0.14 | OOM | OOM
> > > | T1+vp     | 0.012 | 0.021  | 0.026    | 0.035 | 0.075
> > >
> > > In addition to generalizing as well as FNOs and preserving its predictive performance advantages, T1 is much faster and requires less GPU memory to process higher resolution data. For example, FNOs go out of memory on a single GPU with 24GBs of memory when attempting to process a batch of $20$ simulations of resolution $512$, whereas T1 does not. Peak memory usage for T1+ at resolution $2056$ is $7$GBs, while FNOs already peak at $7$GBs while processing data of resolution $256$.
> > >
> > > We would also like to share an interesting insight; working directly in k-space gives T1 models an additional advantage over other FDMs. The T1 outputs in k-space can be reutilized to produce n-space solutions at different resolutions by applying an appropriately-padded inverse DCT, effectively amortizing the cost of the single T1 forward pass. Conversely, FNOs require a different forward pass for each solution at a different resolution. We note that we did not exploit this advantage of T1 while collecting the measurements above: we performed a forward pass of T1s and FNOs for data of each resolution.
> > >
> > > > The authors do not address limitations of their work. This should be fixed if the draft is to published. Conversations about resolution independence, choice of integral transform, and restrictions to regular grids would be welcomed
> > >
> > > We are planning the following concrete steps to address these points in the main text and in an extended limitation Section:
> > >
> > > * Include results on generalization over resolutions discussed above. T1 zero-shot generalize to resolutions better than FNOs, and do so with a drastically smaller memory requirement due to their ability to predict directly in k-space.
> > > * Extend Section 3.1 and the motivation behind DCT-II, clarifying the misunderstanding stemming from our claim of parameter-inefficiency in FDMs with complex transforms (echoing our discussion above). We will include an additional short discussion on limitations of reduced-order T1 models that do not choose the most significant $m$ modes following Section 3.2 but instead truncate at $m$ (smoothing on predictions...).

---

> > > > ### Comment · Reviewer_42JU · 2022-08-08
> > > > **Response to rebuttal**
> > > >
> > > > Thanks to the authors for the rebuttal.
> > > >
> > > > This answers many of my questions and requests that I laid out in the initial review cycle. I still think this is a good submission and still suggest it be accepted for publication. The main advantage of this paper is for me conceptual. I think the idea is worth sharing with the community and I think the authors have done enough to demonstrate (at least in rough form) that the idea of computing everything in the spectral domain is worth further investivation.

---

> > > > > ### Author Response · Authors · 2022-08-09
> > > > > **Thank you for the feedback**
> > > > >
> > > > > Thank you for the response and valuable feedback. Please let us know if there are further questions to address.

---

### Official Review · Reviewer_Pxxo · 2022-07-08

**Rating:** 6
**Confidence:** 4
**Soundness:** 2 fair
**Presentation:** 3 good
**Contribution:** 3 good

**Summary:**

This paper proposes an efficient operator learning scheme by transforming the input data only once to some frequency domain. Numerical results on turbulence simulation indicate its significant speedup and accuracy improvement in several benchmark problems.

**Questions:**

How do you define the architecture of the proposed model in (3)? Also it is not very clear how to define such an architecture which respects symmetry properties of the data, e.g. translation invariance in Navier-Stokes equations.

What are the operations f_1,…f_d in T1(x)? When you compute the speedup, are you computing the same function defined by the FNO? It seems to me there is some issue regarding the speedup results in Fig 3.1, if they were convolutional (e.g. in Unets) as it depends on the filter size.

What is the definition of N-MSE in Table 4.1?

When you talk about the sparsity based on Fourier transforms in the introduction, are you considering turbulence as natural signals? Are they really sparse under DCT transforms?


**Ethics Review Area:**

["I don’t know"]

**Strengths And Weaknesses:**

The idea of computing convolution in the Fourier domain was proposed to speed-up the computational time with respect to the Mathieu et al. 2013, this paper shows that there is some overhead by doing so for each convolution operation: it is more direct to learn some equivalent function in the DCT Frequency domain (section 3.1), without using the common FDM layer structure (eq. 1). The strength seems to be its speedup. A reduced order T1 model is then proposed in 3.2 to further restrict the frequency domain. This seems to be a weakness as one may want to model some energy cascade phenomena in turbulence which would require the whole frequency information in the input signal.

---

> ### Author Response · Authors · 2022-08-02
> **Discussion with Reviewer Pxxo**
>
> Thank you for the review! We add details on how we obtained our speedup measurements and extend our discussion on "sparsity" under DCT. We are also happy to share a variety of interesting additional results, including generalization of T1 to different resolutions (please see our response below to Reviewer 42JU).
>
> > Q: How do you define the architecture of the proposed model in (3)? Also it is not very clear how to define such an architecture which respects symmetry properties of the data
>
> The diagram in (3) shows each component of a reduced-order T1 architecture. First, an input signal $x$ is mapped to the frequency domain via a DCT, yielding $X$. Then, a selection matrix $S_m$ picks the $m$ most-significant spectral domain modes based on a measure of "task-specific importance" (Section 3.2, "Irreducible losses"). Alternatively, $S_m$ selects the lowest $m$ modes, effectively truncating the spectrum.
> A free-form deep architecture $\gamma_\theta$ -- a UNet for T1+ -- is then applied to $S_m X$ to produce the frequency domain coefficients of the output signal $\hat Y$. The loss $J(\hat Y, Y)$ is computed directly in frequency domain. If necessary, desired invariances can be baked in the architecture $\gamma_\theta$, since it is free-form (care should be taken in FDMs, including FNOs, to ensure that the invariance is preserved in signal domain).
>
> > This seems to be a weakness as one may want to model some energy cascade phenomena in turbulence which would require the whole frequency information in the input signal.
>
> We agree, at least in principle. However, our results clearly show that a restricted-order FDM -- that is, an FDM that predicts only $m$ most significant spectral domain modes, produces more accurate predictions in turbulent systems (ScalarFlow), measured by various metrics such as N-MSE and PSNR.
>
> > Q What are the operations f_1,…f_d in T1(x)? When you compute the speedup, are you computing the same function defined by the FNO?
>
> Good question! We use the **exact same** layers $f_1,\dots,f_d$ when computing speedup measurements shown in Figure 3.2. The only difference between T1 and the Fourier Neural Operator here is the removal of intermediate forward and inverse transforms. In particular, we use standard FNO spectral convolution layers ([see code here](https://github.com/zongyi-li/fourier_neural_operator/blob/74b1572d4e02f215728b4aa5bf46374ed7daba06/FNO-torch.1.6/fourier_2d.py#L37)) for both models.
>
> When we mention depth $6$, resolution $128$ and width $32$, we are comparing the following models:
>
> * T1: input signal of resolution $128$ -> DFT -> $6$ layers of spectral convolutions (without forward and inverse transforms) with channel width $32$ -> Inverse DFT
> * FNO:  input signal of resolution $128$ -> $6$ layers of spectral convolutions with channel width $32$
>
> The overhead of forward and inverse transforms grows with higher resolution signals ($O(nlogn)$) and signal width (more transforms to apply in parallel to different channels). Hence the reason for the trend in speedup between T1 and FNOs, which also grows with resolution and width.
>
> > What is the definition of N-MSE in Table 4.1?
>
> We compute the Normalized Mean Squared Error (N-MSE) as
>
> $$
>      \frac{1}{n_y}\sum_{n=1}^{n_y}\frac{(\hat y_n - y_n)^2}{(y_n)^2 + \epsilon}
> $$
>
> where $\epsilon$ is a small positive constant to avoid numerical issues when $y_n = 0$.
>
> > When you talk about the sparsity based on Fourier transforms in the introduction, are you considering turbulence as natural signals?
>
> In the introduction, we refer (informally) to the fact that the quantities dependent on the integral transform of many signals are structured: in particular, they decay in frequency domain with a certain law. For example, the power spectra of natural images often decays as $\frac{1}{k^2}$ where $k$ is the frequency [1]. In some cases, signals might be sparse in the intuitive sense: band-limited simulations such as the Navier-Stokes numerical solutions used in [2] are a notable example. See e.g. Figure 4.1 (right), where the k-space of simulated, incompressible Navier-Stokes has several $\approx 0$ coefficients.
>
> Turbulent systems are certainly not band-limited. Figure 4.3 (right) shows the full k-space of ScalarFlow images: here, decay laws might not be as straightforward as for natural images, but they can still be exploited by a model. As an example, we share a preliminary exploration of an alternative mode selection method. See `lowpass_vs_topk_pass.pdf` in our [anonymous gist](https://gist.github.com/anonymous-conf-sub/9faa4c8e24a13b0e8f78be91b6eef0e7), where we show ScalarFlow reconstruction quality metrics with basic low-pass selection for DCT modes compared to an alternative approach based on the top-k modes with highest energy compaction.
>
> [1] "Modelling the Power Spectra of Natural Images: Statistics and Information", Van der Schaaf et al.,
>
> [2] "Fourier Neural Operator for Parametric Differential Equations", Li et al.,

---

> > ### Comment · Reviewer_Pxxo · 2022-08-03
> > **more questions**
> >
> > Thanks for the detailed reply, they are very helpful. I have some more questions:
> > -  The method FNO proposed by Li et al 2021 has the idea of learning mesh-free, infinite dimensional operators with neural networks, is the proposed method in this paper also sharing the same mesh-free property?  If so, do you have any numerical result about it?
> > - I still think the translational invariance of the operator is very crucial for PDE modeling (to respect the underlying physics), could you give more details about this for the method FNO and also your proposed method?
> > - Besides the training time comparison In Table 4.3,4.4, do you also have any test time comparison (e.g. to see how fast the neural operator works in computing psi(x))? I think this may support the speedup argument that you have made earlier in Section 3.

---

> > > ### Author Response · Authors · 2022-08-03
> > > **Discussion with Reviewer Pxxo**
> > >
> > > Thank you for the responsiveness! Happy to share some new numerical results below to answer your questions.
> > >
> > > > The method FNO proposed by Li et al 2021 has the idea of learning mesh-free, infinite dimensional operators with neural networks, is the proposed method in this paper also sharing the same mesh-free property? If so, do you have any numerical result about it?
> > >
> > > Absolutely! As purely spectral models, T1s are also completely mesh-free. Given a T1 output of DCT coefficients, one can map back to signal domain at any resolution by an appropriately padded inverse DCT.
> > >
> > > We have a completely new experiment on resolution generalization properties of T1s compared to FNOs. We report our comments on the topic, made below to Reviewer 42JU.
> > >
> > > We train FNO and T1+ models on Navier-Stokes data of resolution $64$ x $64$, and test them without finetuning on simulation of different (higher) resolutions:
> > >
> > > N-MSE across different resolutions. Navier-Stokes, viscosity $1e^{-4}$, $T=15$:
> > >
> > > | Method | 64 | 128 | 256 | 512 | 2056
> > > | -------- | -------- | -------- | -------- | ------ |  ------
> > > | FNO     | 0.308     | 0.301     | 0.302 | OOM  | OOM
> > > | T1+vp     | 0.285  | 0.286  | 0.287  | 0.286  | 0.285  |
> > >
> > > Time (s) to process a single test batch of data at different resolutions:
> > >
> > > | Method | 64 | 128 | 256 | 512 | 2056
> > > | -------- | -------- | -------- | -------- | ------ |  ------
> > > | FNO     | 0.049     | 0.049     | 0.14 | OOM | OOM
> > > | T1+vp     | 0.012 | 0.021  | 0.026    | 0.035 | 0.075
> > >
> > > In addition to generalizing as well as FNOs and preserving its predictive performance advantages, T1 is much faster and requires less GPU memory to process higher resolution data. For example, FNOs go out of memory on a single GPU with 24GBs of memory when attempting to process a batch of $20$ simulations of resolution $512$, whereas T1 does not. Peak memory usage for T1+ at resolution $2056$ is $7$GBs, while FNOs already peak at $7$GBs while processing data of resolution $256$.
> > >
> > > We would also like to share an interesting insight; working directly in k-space gives T1 models an additional advantage over other FDMs. The T1 outputs in k-space can be reutilized to produce n-space solutions at different resolutions by applying an appropriately-padded inverse DCT, effectively amortizing the cost of the single T1 forward pass. Conversely, FNOs require a different forward pass for each solution at a different resolution. We note that we did not exploit this advantage of T1 while collecting the measurements above: we performed a forward pass of T1s and FNOs for data of each resolution.
> > >
> > > > Besides the training time comparison In Table 4.3,4.4, do you also have any test time comparison (e.g. to see how fast the neural operator works in computing psi(x))? I think this may support the speedup argument that you have made earlier in Section 3.
> > >
> > > Here are requested wall-clock measurements (same machine and GPUs) from pretrained models. We measure (in milliseconds) average time to perform a single forward pass on the test set of the DFP task (Section 4.2) and the large-scale ScalarFlow experiment (Section 4.3)
> > >
> > > | Method | DFP | ScalarFlow |
> > > | -------- | -------- | -------- |
> > > | FNO     | 211     | 266  |
> > > | T1     | 65 | 67  |
> > >
> > > For additional context, this is the time required for $\gamma_\theta$ in T1: DFP $58$, ScalarFlow $62$. For FNOs: DFP $61$, ScalarFlow $64$ (note that FNOs also perform 1x1 convs between spectral convolution blocks - these have not been counted). The difference between the total wall-clock above and the cost of $\gamma_\theta$ can be interpreted as the cost of spectral domain transforms in each model. As can be seen above, the overhead in FNOs is much greater, leading to T1s being at least 4x faster.
> > >
> > > > I still think the translational invariance of the operator is very crucial for PDE modeling (to respect the underlying physics), could you give more details about this for the method FNO and also your proposed method?
> > >
> > > We agree about the importance of baking inductive biases and invariances in neural models to predict the evolution of dynamical systems. However, we believe this question is outside the scope of our manuscript, since it is not specifically about T1 but rather an open question for all FDMs, including FNOs.
> > >
> > > We would still like to share more details on a potential solution. For the specific case of translational invariance in signal space, one can apply the Fourier shift theorem, which informally states that a shift in signal domain corresponds to multiplication with a complex exponential in spectral domain. Handling these multiplicative factors appropriately can yield a translationally invariant (in signal domain) architecture.

---

> > > > ### Author Response · Authors · 2022-08-09
> > > > **Request for feedback**
> > > >
> > > > As the author-reviewer discussion phase is coming to an end, please let us know if there are further questions or concerns to address.

---

> > > > > ### Comment · Reviewer_Pxxo · 2022-08-09
> > > > > **weak accept**
> > > > >
> > > > > After the discussion, I'd be glad to raise my score to accept the paper. I think the paper quality could still improved by taking into account of all the discussions that we have made so far.

---

> > > > > > ### Author Response · Authors · 2022-08-09
> > > > > > **Discussion with Reviewer Pxxo**
> > > > > >
> > > > > > Thank you for the discussion and for planning to raise the score (small reminder: the OpenReview score has not been updated). The updated manuscript will include additional details that emerged during the discussion phase.

---

### Official Review · Reviewer_kKUQ · 2022-07-11

**Rating:** 3
**Confidence:** 4
**Soundness:** 3 good
**Presentation:** 3 good
**Contribution:** 1 poor

**Summary:**

The authors consider the problem of learning in a frequency domain. They note that multiple Fourier domain layers that perform forward and inverse FFT operations can lead to a high computational cost. They also argue that complex valued layers are inefficient, and propose to use a single forward and inverse DCT throughout the architecture. They also discuss how to initialize such a system.

**Questions:**

- line 94 : Should it be $\mathbb{R}^{m\times n}$ instead of $\mathbb{R}^{n\times m}$?
- in eqn 3 : Is $m$ an optimization parameter -- that is, is $m \in \theta$?
- the eqn between lines 186, 187 : if $m \notin \theta$, minimizers of $J_{\theta}$ and $L_{\theta}$ are the same. $L_{\theta}$ is useful only if you want to compare losses across different values. Is that what you're doing? Are you optimizing over $m$ as well?
- Thm 3.2, line 216, please replace "Then" with "If", and "$\Rightarrow$" with "then". Same comment applies to Corollary 3.1. The statement looks a bit confusing otherwise.

**Strengths And Weaknesses:**

Fourier domain learning is a relatively young but important subfield for machine learning. Therefore the subject is of critical interest. The writing is also clear for most of the manuscript.

However, even though the subject is significant, the contribution to the state of the art is modest. Using a DCT, or any other real-valued harmonic transform instead of the FT is fine, but I'm not really convinced of the argument against using complex valued transforms. If one knows that the signal is real-valued, one can make use of symmetry properties of the FT to avoid any redundant operations. Resorting to the Parseval relation to observe that the error is the same in the transform domain is also perfectly fine, but is a straightforward observation. This leaves a theorem about initialization, as the main contribution. This contribution is welcome, but I think that alone is not sufficient for a Neurips paper.

---

> ### Author Response · Authors · 2022-08-02
> **Discussion with Reviewer kKUQ**
>
> Thank you for the review. We greatly expand on our claim in support of DCT and (hopefully) clear up a minor misunderstanding. The responses below (Reviewer 42JU and Q4Qr) include several additional results: for example, comparing FNOs and T1s on the task of generalization to different resolutions, and evaluating T1-DFTs and T1-DCTs.
>
> > Using a DCT, or any other real-valued harmonic transform instead of the FT is fine, but I'm not really convinced of the argument against using complex valued transforms. If one knows that the signal is real-valued, one can make use of symmetry properties of the FT to avoid any redundant operations
>
> We agree with you that for a real signal $x$, we can put to good use the conjugate symmetry of its Fourier transform $X$, i.e. $X[k] = X^*[N-k]$ can be used to reduce the number of operations required for learned transformation in frequency domain $\hat{X} = f_\theta(X)$. For example, the original Fourier Neural Operator (FNO) [1] ([see here](https://github.com/zongyi-li/fourier_neural_operator/blob/74b1572d4e02f215728b4aa5bf46374ed7daba06/FNO-torch.1.6/fourier_2d.py#L60)) takes full advantage of this symmetry, and we do the same whenever we benchmark against FNOs.
>
> Our argument in favor of real-valued layers however is not based on an inherent computational complexity advantages of applying a single linear layer in real k-spaces. In particular, we observe that training deep models directly in complex k-spaces does not perform as well as in real k-spaces. Similar comments can be found in the neural operator literature (see "Parameterizations of R" , Section 4 [1]).
>
> The empirical observations of DCT outperforming DFT in the tasks we considered could be explained by the sparsity and energy distribution properties of the transformed spaces, i.e. an intrinsic property of the specific dataset and a chosen transform.
> This is in line with results of classic signal processing and compression literature. Particularly, DCT features are known to have a better *energy compaction* (which roughly translates to *sparsity* in ML terms) than their DFT counterparts in a variety of domains, from natural images [2] to audio signals [3] (see e.g. Fig.s 1,2 in the ref.). Energy compaction is often the decisive factor in choosing a transform for downstream tasks e.g., compression (JPEG relies on DCT).
>
> We provide an additional ablation in the Figure `t1_dft_vs_t1_dct.pdf` ([anonymous gist](https://gist.github.com/anonymous-conf-sub/9faa4c8e24a13b0e8f78be91b6eef0e7)). Here, we train a deep architecture in the k-space of a DFT (T1-DFT) and an equivalent model using a DCT-II (the standard T1 model). T1-DFT layers exploit conjugate symmetry of DFT k-spaces, similarly to FNO spectral convolutions: this is the only difference between T1-DCT and T1-DFT, beyond the choice of transform. T1-DFT is less stable during training and diverges, with gradient norms exploding after few training iterations.
>
> This leaves complex k-space FDMs to rely on single (or few) linear layers which are parameter-inefficient when compared against optimized, free-form architectures. Please see `parameters_fno_vs_t1.pdf` in this [anonymous gist](https://gist.github.com/anonymous-conf-sub/9faa4c8e24a13b0e8f78be91b6eef0e7) where we provide a full table of parameter counts at different number of modes $m$ and channel width $c$. For a fair comparison, one should evaluate parameter counts at $m$ for T1+ and $\approx m / 2$ for FNOs due to conjugacy arguments of DFTs on real signals.
>
> There are many more factors that determine which transform is best suited to a task, other than rough computational complexity considerations. We hope the above provides some useful additional motivation behind our specific choice!
>
> > Q: Should it be $\mathbb{R}^{m\times n}$?
>
> Thank you for spotting this typo! $S_m \in \mathbb{R}^{m\times n}$ is correct.
>
> > Is m an optimization parameter. the eqn between lines 186, 187
>
> The number of modes $m$ is not optimized for directly during model training. We do however propose to pick the $m$ best k-space elements for a given task, with Section 3.2 providing one blueprint for selection of these $m$ modes (not necessarily the $m$ lowest frequencies!) for reconstruction or prediction tasks.
>
> > Minor changes to logic operators in Thm 3.2
>
> Thank you for the suggestion; we will introduce these changes.
>
> [1] Li, Zongyi, et al. "Fourier neural operator for parametric partial differential equations." arXiv preprint arXiv:2010.08895 (2020).
> APA
>
> [2] Yaroslavsky, Leonid P. "Fast transforms in image processing: compression, restoration, and resampling." Advances in Electrical Engineering 2014 (2014).
>
> [3] Soon, Yann, Soo Ngee Koh, and Chai Kiat Yeo. "Noisy speech enhancement using discrete cosine transform." Speech communication 24.3 (1998): 249-257.

---

> > ### Author Response · Authors · 2022-08-09
> > **Request for feedback**
> >
> > As the author-reviewer discussion phase is coming to an end, please let us know if there are further questions or concerns to address.

---

> > > ### Comment · Reviewer_kKUQ · 2022-08-09
> > > **Diferences between DCT and DFT**
> > >
> > > Thank you for your responses.
> > >
> > > DCT and DFT definitely have their differences, and I do understand you obtained better results with DCT in your experiments. My main criticism is that, apart from them being real/complex valued transforms, they are aiming a similar (linear) frequency decomposition, and do not necessarily lead to radically different decompositions, if you deal with the boundary issues. To be more precise, *block* DCT performs better on images in terms of obtaining a sparse decomposition because it is more suitable for handling piecewise smooth signals. This is thanks to the underlying symmetry assumptions when it comes to extending the signal -- block DFT would have a hard time with such signals because of the implicit periodic extension (e.g., when repeating the signal leads to a discontinuous periodic signal). However, one could presumably get away with a relatively sparse block-DFT decomposition if one were to apply tapering windows to each block. This is usually done for speech/audio processing when computing the short-time Fourier transform, which is presumably the most popular transform for audio processing. That said, I don't see why one could not use DCT to do audio processing (you've in fact provided an example), because, as I noted, they achieve similar effects -- for each block, they yield a linear decomposition of the frequency axis.
> > > On the other hand, just to give an example, wavelets differ from both block-DFT/block-DCT because the time-frequency decomposition they achieve is quite different, which actually plays very well for images. Continuing the example, wavelets (non-dyadic versions, achieving a higher frequency resolution) can also be used for audio, but are not as popular as the STFT (read block-DFT) presumably because their realization is not suitable for real-time processing.
> > >
> > > To tie it all up, what I would have liked to see in the paper was an argument in favor of transform family X, rather than transform family Y (e.g. wavelets vs block DCT), that outlines their distinct characteristics. In this case, I don't think DCT and DFT really belong to different families.

---

> > > > ### Author Response · Authors · 2022-08-09
> > > > **Discussion with Reviewer kKUQ**
> > > >
> > > > Thank you for the interesting insights. We have provided several arguments based on empirical evidence that outline clear differences between DCTs and DFTs as they relate to the performance of FDMs trained "through" them: one is the experimental section of the paper. The second is provided in the [gist](https://gist.github.com/anonymous-conf-sub/9faa4c8e24a13b0e8f78be91b6eef0e7) (`t1_dft_vs_t1_dct.pdf`), with models trained through DCTs showcasing improved training stability.
> > > >
> > > > It would be helpful to hear the reviewer's opinion on why they think these do not sufficiently justify choosing a transform over the other for learning purposes.
> > > >
> > > > > I do understand you obtained better results with DCT in your experiments. My main criticism is that, apart from them being real/complex valued transforms, they are aiming a similar (linear) frequency decomposition. [...] I don't think DCT and DFT really belong to different families.
> > > >
> > > > We agree that they aim at a similar frequency decomposition. As you mention, there are also measurable experimental differences (for the particular domain of dynamical systems we considered), meaning that taking into account downstream performance DCTs and DFTs do belong to different families.
> > > >
> > > > Both statements are true, and our choice of transform is driven by these experimental differences. We will clarify this point in the main text (Section 3.1). A theoretical analysis and comparison on different transform choices for FDMs is certainly valuable, and a future direction we are planning to explore. However, we believe this is beyond the scope of our work: our focus is to demonstrate that FDMs can be trained directly in frequency domain to improve efficiency while preserving predictive performance over a variety of dynamical systems. In particular, we show that T1s can be trained at a fraction of the computational budget and time (for example, 5 hours versus 32 on ScalarFlow) compared to state-of-the-art FDMs.
> > > >
> > > > Please let us know if there are concerns related to the main message of our work (beyond this particular DFT vs DCT point) that in the reviewer's opinion should be addressed.

---

### Author Response · Authors · 2022-08-02
**Summary of responses**

We thank all reviewers for their detailed and thoughtful comments. We are glad to receive such positive evaluation on all aspects of the submission, including soundness, presentation and contribution.

We provide individual responses to each review. We clarify our claims regarding parameter efficiency of FDMs relying on DCTs versus DFTs, providing parameter counts under varying modes $m$ and model widths, showcasing training instabilities of deep FDMs trained on complex k-spaces (T1-DFT). We further show how our choice is in line with classical signal processing literature, where DCT is often chosen due to its superior energy compaction properties for certain signals.

In addition, we expand our Scalarflow ablation on the vp initialization scheme, which confirms our results of Section 4.1.1. We are also happy to share a completely new experiment on generalization of T1 to different resolutions. T1 is faster and requires less GPU memory when tasked with processing higher resolution data than FNOs at test time.

We believe your comments helped us greatly improve the manuscript. We are happy to answer any additional questions should they arise.

---

### Meta-Review · Area_Chair_b5fC · 2022-08-26

**Recommendation:** Accept
**Confidence:** Certain

**Metareview:**

The paper received three positive reviews (including a well motivated strong accept) and a negative ones. Overall, the feedback provided by the reviewer seem to be useful. The area chair does agree that the concerns of the negative reviewer are a good reason for a reject, even though their review may trigger interesting discussions. The majority of the reviewers believe that the paper makes an interesting contribution and the area chair is happy to recommend an accept.

**Award:**

No

---

### Decision · Program_Chairs · 2022-09-14

Accept